# Synthesis of the Bacteriostatic Poly(l-Lactide) by Using Zinc (II)[(acac)(L)H_2_O] (L = Aminoacid-Based Chelate Ligands) as an Effective ROP Initiator

**DOI:** 10.3390/ijms22136950

**Published:** 2021-06-28

**Authors:** Renata Barczyńska-Felusiak, Małgorzata Pastusiak, Piotr Rychter, Bożena Kaczmarczyk, Michał Sobota, Andrzej Wanic, Anna Kaps, Marzena Jaworska-Kik, Arkadiusz Orchel, Piotr Dobrzyński

**Affiliations:** 1Faculty of Science and Technology, Jan Dlugosz University in Czestochowa, 13/15 Armii Krajowej Av., 42-200 Czestochowa, Poland; r.barczynska-felusiak@ujd.edu.pl (R.B.-F.); p.rychter@ujd.edu.pl (P.R.); 2Centre of Polymer and Carbon Materials, Polish Academy of Sciences, 34 Curie-Sklodowskiej Str., 41-819 Zabrze, Poland; mpastusiak@cmpw-pan.edu.pl (M.P.); bkaczmarczyk@cmpw-pan.edu.pl (B.K.); msobota@cmpw-pan.edu.pl (M.S.); awanic@cmpw-pan.edu.pl (A.W.); 3Department of Biopharmacy, Faculty of Pharmaceutical Sciences in Sosnowiec, Medical University of Silesia, 8 Jedności Str., 41-208 Sosnowiec, Poland; akaps@sum.edu.pl (A.K.); jkik@sum.edu.pl (M.J.-K.); aorchel@sum.edu.pl (A.O.)

**Keywords:** zinc, acetylacetonate, Schiff base, polylactide, antibacterial, bioactive polymer, coordination polymerization, ring opening polymerization

## Abstract

The paper presents a synthesis of poly(l-lactide) with bacteriostatic properties. This polymer was obtained by ring-opening polymerization of the lactide initiated by selected low-toxic zinc complexes, Zn[(acac)(L)H_2_O], where L represents N-(pyridin-4-ylmethylene) tryptophan or N-(2-pyridin-4-ylethylidene) phenylalanine. These complexes were obtained by reaction of Zn[(acac)_2_ H_2_O] and Schiff bases, the products of the condensation of amino acids and 4-pyridinecarboxaldehyde. The composition, structure, and geometry of the synthesized complexes were determined by NMR and FTIR spectroscopy, elemental analysis, and molecular modeling. Both complexes showed the geometry of a distorted trigonal bipyramid. The antibacterial and antifungal activities of both complexes were found to be much stronger than those of the primary Schiff bases. The present study showed a higher efficiency of polymerization when initiated by the obtained zinc complexes than when initiated by the zinc(II) acetylacetonate complex. The synthesized polylactide showed antibacterial properties, especially the product obtained by polymerization initiated by a zinc(II) complex with a ligand based on l-phenylalanine. The polylactide showed a particularly strong antimicrobial effect against *Pseudomonas aeruginosa, Staphylococcus aureus,* and *Aspergillus brasiliensis*. At the same time, this polymer does not exhibit fibroblast cytotoxicity.

## 1. Introduction

Currently, during the global pandemic caused by the SARS-CoV-2 virus, another increasing threat—drug-resistant bacteria—cannot be overlooked [1,2]. In this regard, one of the approaches to mitigate this threat is to search and develop new materials with antibacterial properties.

Many biologically active compounds used as drugs possess modified pharmacological and toxicological potentials when administered as metal-based compounds [3]. Because metal ions or metal ion-binding components play important roles in many biological processes, metal complexes (metallodrugs) are excellent alternatives for treating diseases caused by bacteria. Substances used for antibacterial treatment have low total toxicity; therefore, metals with relatively low toxicity should be considered for developing such substances [4]. Hence, complexes of zinc are one of the best candidates to meet the criteria of low toxicity. Zinc is a bio-metal with a content of approximately 3 g in the human body [5]. Because zinc shows low toxicity [6] and simultaneously strong antibacterial properties, many studies have been conducted on using compounds of this metal for biomedical applications. Among these compounds, zinc complexes containing Schiff bases in the form of chelate ligands show promising potential [7,8]. Schiff bases are products of condensation of primary amines and carbonyl compounds. A comprehensive study of numerous Schiff bases has proved their broad antibacterial and antifungal activities [9]. The most active Schiff bases inhibit bacterial proliferation at concentrations lower than 0.1 mg/mL [10]. A much stronger antibacterial activity is shown by a wide spectrum of metallic complexes containing Schiff bases as ligands [11,12]. The widespread use of these compounds as classical antibiotics for medical treatment, however, remains difficult. It is much easier to use these complexes as antibacterial additives for objects of everyday use or for medical devices and dressings, such as polymers, plastics, fibers, fabrics, biodegradable implants, sutures, and wound dressings. These materials play an important role in food packaging, but they are mostly used for controlling infection caused by implanted biomaterials in medical applications [13,14,15,16]. 

Implants and medical products made from bioresorbable polymers, mostly aliphatic polyesters such as lactide copolymers, are currently attracting great interest in the field of medicine. These polymers are well known because of their good properties, such as biocompatibility and bioresorption, and products made from these polymers are certified as safe for human use. To maintain the bioresorbable and biocompatible properties of these products, the use of polymeric biologically active additives with defined biocompatibility is a much easier and faster approach than chemical modification of the polymer or application of new functional monomers with bioactive side groups. Therefore, the most common polymer systems exhibit antibacterial properties through the formation of a polymer composite containing a dispersed bioactive agent in its mass [17,18]. However, it is difficult to ensure proper homogeneity of the final product prepared in this manner because, frequently, the bioactive substance is present in the form of different types of conglomerates or crystallites. Moreover, this method is relatively difficult and technically inconvenient. In the present study, we show a different approach to this goal by using zinc complexes with proven antibacterial activity as a lactide ring-opening polymerization (ROP) initiator at the stage of polymer synthesis. This method allowed to obtain biologically active polymeric composites with a uniform morphology and containing an antibacterial complex in the dispersed molecular form, which is practically impossible to obtain through physical methods. 

## 2. Results and Discussion

### 2.1. Synthesis of Schiff Bases 

After reviewing the literature and based on the results of our preliminary research, a group of complexes expected to possess the desired properties was selected. The complexes should show good catalytic activity during the initiation of lactide ROP. In this regard, zinc complexes were considered to fully meet the expected requirements, i.e., show robust antibacterial activity and relatively low toxicity [19,20,21].

Antibacterial and antifungal activities of zinc (II) acetylacetonate (Table 1, Row 1) were comparable to those of endogenous amines (Table 1, Rows 2–3). The level of activity of these amines, estimated during our tests, was very similar to the results obtained earlier [22,23]. This zinc complex is also an effective initiator in ROP of lactide [24] or trimethylene carbonate [25]. These results agree with other reports, showing good catalytic properties of the ROP of the zinc (II) chelate complexes, including complexes with Schiff base ligands [26]. In our study, Schiff bases synthesized by condensation of selected amino acids were chosen because of their relatively low toxicity and strong antibacterial effect [27,28]. To obtain the desired antibacterial activity of the compounds, the pyridine derivative was selected as a second substrate used during condensation [29]. Finally, in the study, we used the Schiff bases HTrp and HPhe obtained through condensation of tryptophan (Trp) or l-phenylalanine (Phe) with 4-pyridinecarboxaldehyde (Figure 1). 

The composition of the synthesized Schiff bases was confirmed by ^1^ H NMR, FTIR, and elemental analysis. In the ^1^H NMR spectrum of HPhe (Figure 2a), a lack of signals assigned to the proton of aldehyde group of pyridinecarboxaldehyde (CH–OH at 10.2 ppm) and protons of the -CH_2_–CH–NH_2_ group at 3.4 ppm assigned to phenylalanine was observed. Instead of that, new signals appeared: signal 7 of group =N–CH- at 3.9 ppm, and 8 of group =N–CH–CH_2_- at 3.0 and 3.3 ppm (in the spectrum of unreacted phenylalanine signals, -CH_2_–CH–NH_2_ appeared at 2.8 and 3.1 ppm). 

In the FTIR spectrum of LPhe (Figure 3a), the bands at 1705 and 1625 cm^−1^, corresponding to the stretching vibrations of the C=O aldehyde groups and deformation vibrations of the NH_2_ phenylalanine groups, respectively, were not detected, which indicates the occurrence of a reaction between the aldehyde and phenylalanine. The formation of imine was confirmed by the band at 1647 cm^−1^, originating from the stretching vibrations of the C=N group. Changes observed in the region of 1610–1500 cm^−1^ resulted from a change in the charge distribution of the aromatic rings due to C=N conjugation [30].

The results of the elemental analysis of HPhe were also similar to the theoretical composition (in brackets): C—69.3 (70.9); H—5.3 (5.6); and N—10.1 (11.0). 

The synthesis of HTrp through condensation of tryptophan has already been described earlier; however, ^1^H NMR analysis was performed for the first time in detail (Figure 2b). Analogous to the spectrum discussed earlier, Signals 1, 7, and 8, assigned to the characteristic groups of protons of HTrp, appeared practically at the same values of the chemical shifts (7.4, 3.85, 3.1, and 3.15 ppm). There was also a lack of signal characteristic for the protons in the neighborhood of the amine group of tryptophan -CH–CH_2_–NH_2_ (3.48, 3.0, and 3.35 ppm, respectively) as well as signals assigned to the proton of the aldehyde group of pyridine carboxaldehyde. 

In the FTIR spectrum of LTrp (Figure 3b), the bands attributed to the stretching vibrations of the C=O aldehyde groups (1705 cm^−1^) and deformation vibrations of the NH_2_ tryptophan groups (1666 cm^−1^) were also not observed. In this case, the band characteristic for the stretching vibrations of the C=N groups appeared at 1642 cm^−1^, thus confirming the formation of imine structure. 

The results of the elemental analysis of HTrp were very close to the theoretical composition (in brackets): C—68.3 (69.6); H—5.9 (5.2); and N—12.9 (14.3).

Assessment of the antibacterial and antifungal activities of both compounds (HPhe and HTrp) (Table 1, Rows 5 and 6) showed higher activity than that previously reported for Schiff bases obtained in the condensation of the same amino acids but with other aldehydes, such as 3-methoxysalicylaldehyde or 4-diethylaminosalicylaldehyde [28]. The synthesized Schiff bases at the concentration of approximately 10 mg/mL showed the capability of inhibiting the growth of most tested microorganisms, which is comparable to the activity of diamines, such as spermidine or putrescine (Table 1, Rows 2 and 3). Both HTrp and HPhe showed high activity against *Staphylococcus epidermidis* and *Aspergillus brasiliensis*, and the concentration of approximately 1 mg/mL was sufficient to decrease the growth by about 90%. Similar Schiff bases obtained using the same amino acids but by condensation with salicylaldehydes exhibited effective activity at a concentration close to 30 mg/mL [28].

### 2.2. Synthesis and Structure of Zinc(II) Complexes with Schiff Bases and Acetylacetonate Ligands

Both the obtained Schiff bases were used for further synthesis of new zinc complexes conducted by the ligand exchange reaction of the Zn[(acac)_2_ H_2_O] complex. Although the reaction was conducted using various molar ratios of zinc(II) acetylacetonate to the Schiff base (as 1:1, 1:2, and 1:3), the resultant complexes showed the same final composition with an equimolar amount of ligands.

The structure and composition of the obtained complexes were estimated by NMR and FTIR spectroscopy as well as by elemental analysis. Figure 4, shows the ^1^H NMR spectrum of the complexes. In the spectrum of both complexes, two intensive signals of the acetylacetonate ligand groups CH_3_ and CH were observed. Compared to signals of the analogous acac groups of initial Zn[(acac)_2_ H_2_O] measured in DMSO d6 (CH—5.27 ppm, CH_3_—1.86 ppm), these signals were found to be shifted. The acetylacetonate signals of Zn[(acac)(LPhe)H_2_O] appeared at 5.13 and 1.7–1.8 ppm, while the signals of Zn[(acac)(LTrp)H_2_O] appeared at 4.64 and 1.7–1.85 ppm. Accurate measurements of the signal intensity demonstrated an equimolar amount of acetylacetonate and Schiff base ligands in both complexes. Comparison of the spectra of the complexes to the Schiff bases outputs showed that the most apparent changes were observed for Signals 1, 7, and 8 (Figure 3.), which are assigned to protons surrounding the azomethine group -CH=N-. In the spectrum of Zn[(acac)(LPhe)H_2_O], the following signals shifts were observed: from 7.45 to 7.4 ppm, from 3.85 to 4.42 ppm, and from 3.0, 3.3 to 2.5, 2.7 ppm, respectively (Figure 4a). Similar shifts in the spectrum of Zn[(acac)(LTrp)H_2_O] were also observed (Figure 4b). The following shifts were observed: Signal 1 from 7.4 to 7.32 ppm, Signal 7 from 3.85 to 3.93 ppm, and Signal 8 from 3.15 and 3.1 ppm to 3.1 and 2.8 ppm. This implies that because of the formation of complexes containing the new ligand, a new bond is formed that links the nitrogen ion pairs from the imine group with the central zinc atom. These phenomena caused a change in electron distribution, which resulted in a chemical shift of the ^1^H NMR spectra signals. Furthermore, the shift in signals and splitting of protons of the aryl groups were observed, especially in the spectrum of Zn[(acac)(Lphe)] (Figure 4a, in the range of 8.7–7.5 ppm). A similar phenomenon was already observed in the spectra of the Pt(II) complexes of the Schiff base derived from l-phenylalanine and furfuraldehyde [31]. The signals assigned to water protons were much more intensive than those of water absorbed in DMSO d6; this most probably indicated that the synthesized complexes also contained a water molecule, similar to that for Zn[(acac)_2_ H_2_O]. 

In the FTIR spectrum (Figure 5) of Zn[(acac)_2_ H_2_O], the most characteristic bands appeared at 1557 and 1447 cm^−1^ and originated from the asymmetrical and symmetrical stretching vibrations of the (C=O)–O¯ groups, respectively. In the complexes with mixed ligands, these bands were shifted to 1512 and 1398 cm^−1^ for Zn[(acac)(LPhe)H_2_O] (Figure 5a) and to 1518 and 1399 cm^−1^ for Zn[(acac)(LTrp)H_2_O] (Figure 5b), thus indicating a charge distribution in the ligands after the addition of the new Schiff base ligand. Additionally, the intensity of the band recorded at 1557 cm^−1^ was strongly diminished, thereby confirming the lower content of acetylacetonate ligands. On the other hand, the bands originating from the C=N group vibrations in the spectra of both Schiff bases disappeared due to the interactions between the C=N group and the Zn atom. The Zn atom caused a shift of the electrons from the C=N group toward Zn, which lowered the double-bond strength of the C=N group [32].

Analysis of the elemental composition of the complexes confirmed the spectroscopic results. The elemental analysis revealed the following values of the elements: Zn[(acac)(Lphe)H_2_O]: C%—55.9 (55.2), H%—5.4 (5.1), and N%—6.1 (6.5); and Zn[(acac)(LTrp)H_2_O]: C%—53.7 (55.7), H%—5.2 (4.9), and N%—8.4 (8.9). The theoretical values of each element are given in the brackets on the assumption that there are zinc complexes with a mononuclear structure containing one acetylacetonate and one Schiff base ligand, as well as one water molecule.

Because the obtained zinc complexes were planned to be used as polymerization initiators, to understand the hypothetical initiation process, it was necessary to appreciate their molecular geometry. The arrangement of the atoms of zinc(II) acetylacetonate monohydrate, which was a precursor of our complexes, was investigated previously in the 1960s [33]. It was found that the molecular geometry of such a type of complex could be estimated by quantum mechanical studies, yielding high compatibility with the geometry determined experimentally [34]. 

In the present study, the Zn(II) complexes were optimized and analyzed with the Gaussian 03 program using the DFM method, with a basis set in the gas phase. Both complexes showed a similar geometry (Figure 6). Based on the calculations, we consider that the Zn(II) complexes had a distorted trigonal bipyramidal geometry. This type of structure for the Zn(II) complex was first described in 1995 [35]. 

Zn[(acac)(LPhe)H_2_O] showed such a strong deformation that we can safely consider this geometry as the transition between the square pyramidal structure (typical for zinc(II) acetylacetonate monohydrate) and the trigonal bipyramidal structure (Figure 6a). For the second complex (Figure 6b), the deformation was not that strong. The calculated lengths of Zn-O1 and Zn-O2 and the angle O1-Zn-O2 (acetylacetonate ligand) in Zn[(acac)(LPhe)H_2_O] were, respectively, 2.019 Å, 2.013 Å, and 99.75°; those for Zn-O3 and Zn-N and the angle O3-Zn-N (Schiff base bonds) were, respectively, 2.020 Å, 2.182 Å, and 84.06°; and that for Zn-O4 was 2.60 Å (water molecule). For Zn[(acac)(LTrp)H_2_O], the lengths of the bonds were as follows: Zn-O1—1.970 Å, ZnO2—1.968 Å, Zn-O3—1.982 Å, Zn-N—2.121 Å, and Zn-O4—2.58 Å; and the angles were as follows: O1-Zn-O2—105.51° and O3-Zn-N—81.96°. The calculated geometry was identical to that of many other Zn(II) mononuclear complexes with a similar geometry; for example, Zn(4-dimethylaminopyridyl)(acac)_2_] [36] or Zn[(PhCOO)(cur)(bpy)] contained ligands on curcumin, bipyridine, and benzocarboxylate [37]. The respective lengths of the bonds between zinc and the oxygen or nitrogen atoms as well as the angles in these complexes were very close to the values presented above. For Zn [(PhCOO) (cur)(bpy)], the values of these parameters were as follows: Zn-O1 and Zn-O2—2.01 Å and 2.05 Å; Zn-N1 and Zn-N2—2.11 Å and 2.13 Å; angle O-Zn-O—89.8 °; and N-Zn-O—90.5 °.

### 2.3. Antibacterial and Antifungal Activities of the Obtained Zn(II) Complexes

The synthesized zinc complexes were subjected to analogous tests for antibacterial and antifungal activities, similar to that for the previously reported Schiff bases. The obtained results are shown in Table 1 (Rows 7–8). Compared to the used Schiff bases and the original Zn[(acac)_2_ H_2_O], a very distinct increase in the antimicrobial activity was observed against all strains selected in the test. Zn[(acac)(LTrp)(H_2_O] was particularly active against Escherichia coli, S. epidermidis, and Candida albicans. Significant growth retardation of these strains by the complex was already observed at a concentration lower than 0.1 mg/mL. Zn[(acac)(LPhe)(H_2_O] also inhibited the growth of Pseudomonas aeruginosa, S. epidermidis, and Staphylococcus aureus at identical concentrations. Furthermore, the higher concentrations of 1 mg/mL and 10 mg/mL resulted in a rapid and complete inhibition of the growth of P. aeruginosa and the other strains, respectively. This relatively high antimicrobial activity and, in particular, the broad spectrum of action of this compound, make this complex very interesting in terms of its intended use as a universal antibacterial additive for polymer materials. Comparison of the obtained results with those of similar previously published studies further confirmed the relatively high antibacterial activity of these complexes. For example, Zn(II) complexes with low-toxic Schiff base ligands, obtained based on l-tryptophan and dihydroxybenzaldehyde, showed activity only against E. coli [38]. For Fe(II) hydroxylnapthylidene amino acid complexes, this compound showed a similar activity to our prepared complexes against P. aeruginosa and E. coli but at higher concentrations, i.e., approximately 10–20 mg/mL. However, these complexes inhibited fungal growth only to a small extent [39]. Our obtained zinc complexes exhibited much better biological activity than the initial Schiff bases. This observed increase in the antibacterial activity of the complexes relative to the starting organic ligands was also reported previously for many compounds of this kind. This phenomenon can be explained by the synergistic effect between zinc and the organic ligands of the complex, resulting in a strong increase in the penetration of the bacterial lipid membrane [40]. It is also possible to obtain compounds with stronger antibacterial activity than the zinc (II) complexes obtained by us. For example, zinc chelate complexes containing special ligands synthesized by reacting acetamido benzaldehyde with 2-amino-6-methylsulfonyl-benzothiazoles showed a very strong antibacterial effect against three selected strains of bacteria at a concentration of only approximately 0.03 mg/mL [8]. However, because of the presence of benzothiazole derivatives, the complex shows relatively high toxicity, which was avoided in our present study.

### 2.4. Lactide Polymerization Using the Obtained Zinc Complexes as a Coordination Initiator

The next step of our study was to determine the efficiency of the synthesized complexes as initiators in ROP of lactide. For this purpose, a few attempts of l-lactide polymerization using both complexes were conducted in bulk at 110 °C, with the initiator-to-monomer (I/M) molar ratios of 1:150, 1:400, and 1:600 (Figure 7). For comparison, an analogous polymerization under the same conditions was conducted using Zn[(acac)_2_H_2_O]. Both new complexes, regardless of the concentration, easily dissolved in melted l-lactide and formed a homogeneous phase with the melted monomer, and both of them were found to be effective initiators. However, the Zn[(acac)(LTrp)H_2_O] complex showed slightly higher activity. In the reaction conditions, by using this initiator, the conversion of the monomer ranged between 80% and 90% within 24 h (Figure 7b). Monomer conversion during the use of the second complex Zn[(LPhe)(acac)H_2_O] was slightly lower and ranged between 70% to almost 90% in 24 h (Figure 7a). Both complexes were found to be much more active as initiators in the ROP of lactide as compared to the previously described zinc acetylacetonate (Figure 7a, Serie 4), which demonstrated a conversion close to 90%, but only after approximately 70 h [24]. 

As the lactide conversion increased, the average molecular weight of the resulting polymer also increased throughout the reaction time. We observed a linear relationship up to approximately 60–70% lactide conversion (Figure 8), which corresponded to approximately 10–12 h of polymerization. 

We observed an increase in the difference between the values of the average molecular weight calculated and measured along the reaction time. This probably occurred due to the effect of thermal degradation of the synthesized polymer [41]. The observed relationship between the amount of initiator and the average molecular weight of polylactide confirmed that both the examined complexes possess mononuclear structures. 

### 2.5. Mechanism of Initiation of Lactide ROP Through Involvement of the Obtained Zinc Complexes

Differences in the initiation mechanism of lactide polymerization were the reason for the observed much greater effectiveness of the zinc complexes containing ligands based on amino acids as compared to zinc acetylacetonate. In previous papers, we demonstrated the application of zinc(II) and zirconium(IV) acetylacetonates as an initiator of lactide polymerization or copolymerization [24,41,42,43,44]. The first step of this reaction is the formation of the active initiating complexes through an exchange reaction of the acac ligand and deprotonated lactide with the release of free acetylacetonate—Hacac. This complex, containing a derivative of lactide as a ligand, is finally an active initiator of polymerization. 

To follow the proceeding changes, lactide polymerization was conducted at a relatively high amount of initiator (molar ratio I/M of 1:12). This model reaction was performed in benzene because this solvent is inert and not influenced by the initiation mechanism (nonpolar solvent without the ability to coordinate to the metal of the central complex). Figure 9 shows the ^1^H NMR spectra of the obtained reaction products. The analysis of the the spectra showed the continuous presence of the CH_3_ signal of the acetylacetonate ligand during the polymerization process. Admittedly, as the polymerization progresses, this signal is split more and more, which is associated with the formation of the polymer chain; however, the average intensity of the signals was constant throughout the investigation. There was also a lack of characteristic signals related to the free acetylacetonate groups, Hacac, which should occur for proton transfer and ligand exchange reactions, as was observed during the initiation of the polymerization of Zn[(acac)H_2_O] [43]. Initially, the signal of the CH group of the acetylacetonate ligand was also visible, but with the progression of reaction time, it was overshadowed by the CH signals of the growing polylactide chain. This observation proves that the acac ligand was coordinated with the central atom of the initiating complex throughout the studied polymerization process. Thus, the observed initiation polymerization mechanism with zinc complexes containing LTrp and LPhen ligands is different from that previously described for Zn[(acac)_2_ H_2_O] or Zr(acac)_4_. In that case, the Schiff base ligands must play an essential role in the monomer insertion process. This assumption might be proved by the observation of Signal 7, which was present throughout the entire reaction time and shifted strongly to 4.72 ppm (it was 4.42 ppm in the initial complex and 3.85 ppm for the Schiff base). Signal 8b—characteristic for the protons of the Schiff base ligand in the initiating complex (see also Figure 4a, Signal 8 at 2.5–2.7 ppm)—disappeared as the polymerization progressed (Figure 9, Signal 8b—occurring in Zn [(acac)(LPhe)H_2_O]). This signal also appeared in the spectra, initially mainly as Signal 8a shifted to 3.05 and 3.15 ppm and then as a rising signal at 2.77 and 2.95 ppm (Figure 9I, Signal 8a and 8c). The structure of the initiating complex then changed, and most probably, the derivative of the Schiff base was continuously linked to the polylactide growing chain, as illustrated in the enclosed scheme (Figure 9II, a,b). These observations suggest that the mechanism of initiating lactide polymerization with Zn[(acac)(LPhen)] observed in this work is analogous to the previously described mechanism of initiation of polymerization of trimethylene carbonate (TMC) with the complex obtained in the reaction of Zr(acac)_4_ and benzoic acid. [43]. The ROP initiating complexes described in the current work are similar in some aspects to those discussed in previous work. They were also obtained by an acetylacetonate ligand exchange reaction, in this case with a Schiff base. However, the full confirmation of this hypothetical mechanism requires many additional studies, which is beyond the main scope of this work. 

### 2.6. Bactericidal and Fungicidal Properties of Polylactide Obtained through Polymerization Initiated by Zn [(acac)(LPhe)H_2_O] and Zn[(acac)(LTrp)H_2_O]

By using the obtained zinc complexes as a polymerization initiator, a series of poly(l-lactide) samples was obtained. The polymers were prepared with various amounts of initiator (with I/M molar ratios of 1:150 and 1:400). This polymer, after grinding and sieving (a fraction with a diameter of 10 to 100 µm) in the form of an aqueous suspension, was subjected to antimicrobial tests to determine its antibacterial activity. Poly(l-lactide) obtained with Tin(II) 2-ethylhexanoate (a typical initiator used industrially in the production of this polymer) was used as a comparative sample. The biological activity of the samples was tested after 24 and 48 h of incubation. Table 2 presents the results of the tests.

Surprisingly, the tested polylactide exhibited antibacterial activity similar to the complexes used as the initiator of polymerization. The polymeric materials obtained with Zn[(acac)(LPhe)H_2_O] (Table 2, Rows 2 and 3) also showed interesting results. The polymer obtained with a higher amount of initiator (I/M = 1:150, Mn at approximately 10,000 g/mol) showed the highest activity against most of the tested strains. However, this material, even at high concentrations, showed minimal ability to inhibit the growth of *C. albicans*. In this case, the antibacterial activity was even lower than that observed for a reference sample of polylactide obtained using Tin(II) 2-ethylhexanoate initiator (Table 2, row 1). On the other hand, polymer obtained with Zn[(acac)(LPhe)H_2_O] was particularly active against *E. coli*, where the polymer at 1 mg/mL concentration was sufficient to completely inhibit the cell growth of this strain. The material was also very active against *S. epidermidis*, *S. aureus*, and *A. brasiliensis*. The polymer obtained with a lower amount of Zn[(acac)(LPhe)H_2_O] (I/M ratio as 1:400, Mn = 31,700 g/mol) showed lower antibacterial activity against most of the tested strains (Table 2, row 3). However, it still showed significant activity against bacteria, especially after 48 h of incubation. The polymer obtained in the polymerization initiated with Zn[(acac)(LTrp)H_2_O] exhibited significantly lower antibacterial activity, but showed better activity against *C. albicans*, which was the extremely resistant strain in most of our tests (Table 2, row 4).

The antibacterial properties of the synthesized polylactide were evident. Given that the content of the zinc complex of the polymer was approximately 1.9 wt.% of the entire mass of the polymer, its activity against the tested microorganisms appeared to be strong. This phenomenon is difficult to explain without additional research that could confirm the possibility of a synergistic effect between the initiator residues and degradation products as well as with the remnants of unreacted lactide. However, such detailed research will be the subject of a new research paper. 

### 2.7. Cytocompatibility of Polylactides Obtained by Using Zn[(acac)(LPhe)H_2_O] and Zn[(acac)(LTrp)H_2_O] as an Initiator

To estimate the degree of the potential toxicity of the obtained polymers, tests of their cytotoxicity against fibroblast lines were carried out. The assessment was performed on crude polylactide samples obtained with both initiators (with an M/I molar ratio of 1:400) and at different monomer conversion rates. Figure 10A shows the viability of CCD-11Lu fibroblasts treated with extracts from polymers obtained with Zn[(acac)(LTrp)H_2_O] with 85% and 96% lactide conversion, assessed employing the resazurin reduction assay. Cytotoxicity profiles of both materials were similar, regardless of the conversion rate. Undiluted extracts caused almost complete suppression of cellular growth, similarly to the 1:2 dilution. Extracts diluted 1:4 caused only partial inhibition of cell growth, whereas larger dilutions did not significantly affect cellular proliferation. As shown in Figure 10B, polylactide initiated with Zn[(acac)(LPhe)H_2_O] displayed excellent biocompatibility but only after a lactide conversion rate of 93%. In the case of that material, even the undiluted extract did not cause any decrease in cell viability. Simultaneously, the material characterized by the lower conversion rate had a cytotoxicity profile very similar to the above-described polymers. We do not present results of LDH release assays for polymeric extracts because some extracts apparently inhibited the activity of lactate dehydrogenase, and hence, in this case, it could not be considered as a reliable indicator of cell viability. 

The cytotoxicity of the initiator itself was also assessed. Figure 11 shows the results of cytotoxicity tests of Zn[(acac)(LPhe)H_2_O]. Both the used techniques, namely, the resazurin reduction and LDH release assays, unanimously proved that the tested compound decreased the cell viability exclusively at a relatively high concentration of 1 mg/mL. This means that in concentrations showing a strong bactericidal effect, this compound displays practically no cytotoxicity. Unfortunately, we were unable to perform an analogous assay for the Zn[(acac)(LTrp)H_2_O] initiator because of its too poor water solubility (as it formed a suspension, sterilization with the use of a membrane filter was impossible). 

## 3. Materials and Methods

### 3.1. Materials

Zinc (II) acetylacetonate monohydrate (Alfa Aesar, Ward Hill, MA, USA), l-tryptophan, l-phenylalanine, 4-pyridinecarboxaldehyde, methanol anhydrous 99.8%, benzene anhydrous 99.8%, chloroform anhydrous 99%, and tetrahydrofuran anhydrous 99.9% were purchased from Sigma-Aldrich, Poland, and potassium hydroxide reagent grade was received from Merck. All these chemicals were used as received. 

l-lactide (Forusorb, medical grade) was received from Foryou Medical Device Co., Ltd. Huizhou, Guangdong, China, and before use was purified by recrystallization from dry ethyl acetate. 

### 3.2. General Procedure for the Synthesis of the Schiff Base Ligands HPhe and HTrp

The Schiff bases were synthesized using a previously reported method, with some modification [29,45]. The reaction was performed in an anhydrous methanol solution under an argon blanket in an alkaline medium (0.05 mmol KOH/mL). After dissolving 10 mmol of phenylalanine in 100 mL methanol solution, a stoichiometric amount of 4-pyridine carboxaldehyde was added dropwise with constant stirring; the temperature of the solution was gradually increased to 50 °C, and the solution was stirred for 2 h. After lowering the solution temperature to approximately 30 °C, the process was continued for the next 24 h. The solution was then highly concentrated on an evaporator (evaporation volume approximately 15–20% of the original solution volume) and allowed to crystallize. Yellow fine crystals were obtained, which were washed with methanol and recrystallized again from methanol. The product was obtained with approximately 67% yield.

The second compound was prepared analogously by reacting 10 mmol l-tryptophan and 10 mmol 4-pyridinecarboxaldehyde. After the solution was concentrated, approximately 20 mL of anhydrous butyl ethyl ether was added, resulting in the formation of a yellow-orange thick oil at the bottom of the vessel. After 24 h storage at approximately 10 °C, the oil crystallized into orange crystals. After separating the crystals, the product was washed with ether and recrystallized from methanol. The yield was approximately 51%.

### 3.3. General Procedure for the Synthesis of Zinc Complexes; Zn[(acac)(LPhe)H_2_O] and Zn[(acac)(LTrp) H_2_O]

The previously prepared Schiff bases were used in the reaction with zinc (II) acetylacetonate monohydrate. The reaction was performed in 150 mL glass reactors, which were heated and equipped with a magnetic stirrer, a reflux condenser, and an argon gas supplier. The method of synthesis used was a modification of the previously published method describing the preparation of cobalt, magnesium, and manganese complexes containing acetylacetonate ligands and Schiff bases [46]. 

Approximately 100 mL of anhydrous methanol was placed in the reactor and heated to a temperature of about 60 °C. Next, Zn[(acac)_2_ H_2_O] (3 mmol) was dissolved in methanol, and with continuous stirring, methanolic solutions of HPhe or HTrp were gradually added dropwise in a zinc acetylacetonate/Schiff base molar ratio of 1:1, 1:2, or 1:4. After the dropwise addition, the solution was stirred under reflux at 60 °C for the first 4 h and 40 °C for the next 20 h. The color intensity of the reaction solution was markedly increased. The reaction solution was then concentrated on an evaporator (to approximately 20% of the original volume). After cooling for a few hours, a dark yellow (Zn[(acac)(LPhe)]) or dark orange (Zn[(acac)(LTrp)]) fine crystalline precipitate was formed, which was centrifuged, washed with methanol, and then dried. However, the obtained single crystals were not suitable for X-ray crystallography. For the reaction performed using the reagents in the stoichiometric ratio of 1:1, the products of the zinc complexes were obtained with a yield of approximately 57% and 68%, respectively.

### 3.4. General Procedure for the Synthesis of Poly(l-Lactide)

The lactide polymerization process was investigated under bulk conditions with different contents of initiators: Zn[(acac)(LTrp)H_2_O] and Zn[(acac)(LPhe)H_2_O] complexes. For this purpose, 2 g (13.9 mmol) of l-lactide was added to a 10 mL glass reactor equipped with a magnetic stirrer and an argon gas supplier, and the monomer was then melted under argon at 120 °C. Measured amounts of the previously obtained zinc complexes (9.27 × 10^−2^ mmol, 3.48 × 10^−2^ mmol, 2.32 × 10^−2^ mmol) were added as solutions in chloroform to the vigorously stirred melt. The reaction was allowed to continue for a planned time and then terminated by rapidly cooling the content. The obtained samples were examined by ^1^H NMR spectroscopy and gel permeation chromatography to determine their composition and average molecular weights.

To understand the initiating stage of lactide polymerization with the obtained complexes, a model oligomerization was performed using a relatively large amount of Zn[(acac)(LTrp)H_2_O]. The reactions were conducted in a benzene solution at 100 °C in a 150 mL glass reactor equipped with a magnetic stirrer, an argon gas supplier, and a reflux condenser. Next, 60 mL of anhydrous benzene was added to the reactor. After heating to 60 °C, 4.17 mmol of Zn[(acac)(LPhe)H_2_O] was introduced into the vessel. Subsequently, when the initiator had dissolved, 50 mmol of l-lactide was added. For the tests, samples were taken with a syringe after 20 min and 1.5 and 9 h of the reaction. 

The polymers for biological tests were obtained by polymerization carried out for 36 h under bulk conditions at 120 °C (Table 3). The polymers were ground using a cryogenic grinder and sieved. The fraction with a diameter of 10 to 100 µm was selected for the subsequent tests.

### 3.5. Estimation of the Antibacterial and Antifungal Activities of the Tested Samples

Inhibitory concentrations were estimated using a microtiter broth dilution method, as recommended by the Clinical and Laboratory Standards Institute [47]. Samples of each tested compound or polymers were prepared at the concentrations of 20, 10, 1, and 0.1 mg/mL by making an aqueous solution (for water-soluble samples) or a water suspension and then quickly tested. Serial two-fold dilutions of the different test compounds were prepared in test tubes. Test tubes without the test compounds were used as the positive growth control. A diluted bacterial suspension was added to each test tube to yield a final concentration of 5 × 10^5^/5 × 10^6^ colony forming units (cfu)/mL, as confirmed by viable cell count (determined by turbidimetric method). Bacterial inoculum was used as the negative growth control. The plates were incubated at 37 °C for 24 and 48 h. The contents of the test tube showing no visible growth were plated on selective substrates, and the number of colonies was counted after overnight incubation at 37 °C. For each strain, at least three independent determinations were performed, and the modal value was taken. The following strains were selected for the study: 

Gram-positive: 

*Staphylococcus aureus* NCTC 10788/ATCC 6538; 

*Staphylococcus epidermidis* NCTC 13360/ATCC 1222.

Gram-negative:

*Escherichia coli* NCTC 12923/ATCC^®^ 8739;

*Pseudomonas aeruginosa* NCTC 12924 / ATCC 9027.

Fungi:

*Aspergillus brasiliensis* NCPF 2275/ATC C 16404.

Yeast: 

*Candida albicans* NCPF 3179/ATCC 10231.

All strains were obtained from Biomaxima S.A., Biocorp Microbiology Center, Poland. The following substrates were used for culture: *E. coli*—MacConkey Agar; *P. aeruginosa*—Cetrimide Agar; *S. aureus* and *S. epidermidis*—Chapman-Mannitol Salt Agar; *C. albicans* and *A. brasiliensis*—Sabouraud Dextrose Agar. All substrates were obtained from Biomaxima S.A., Poland. 

To estimate the MICs of the synthesized compounds, the following concentrations were used: 0.001, 0.01, 0.1, 1, and 10.0 mg/mL.

### 3.6. Cytocompatibility Studies 

In vitro cytocompatibility of both the polymeric materials and initiators was studied using the human normal CCD-11Lu fibroblast cell line (ATCC; CCL-202). Initiators were simply dissolved in a complete culture medium and solutions were filtered through 0.2 μm syringe filters. Polymers were extracted in a complete culture medium according to ISO 10993-12. To prepare the extracts, polymeric specimens were dipped in medium (0.1 g of material per 1 mL of medium) and incubated for 24 h with gentle agitation. Extracts were sterilized by filtration (0.2 µm syringe filters) and diluted 1/1 (undiluted), 1/2, 1/4, 1/8, and 1/16. The control culture medium was treated in the same way as the extraction medium except that there was no polymeric specimen. CCD-11Lu cells were cultured in MEM medium (Minimum Essential Medium Eagle, Sigma-Aldrich) supplemented with 10% FBS (Pan Biotech), 100 U/mL penicillin, 100 µg/mL streptomycin, and 10 mM HEPES (Sigma-Aldrich). Cells were maintained at 37 °C in a humidified atmosphere containing 5% CO_2_. To test for cytotoxicity, cells were seeded into 96-well plates (10^4^ cells/well in 100 μL of culture medium) and allowed to adhere for 24 h. Subsequently, the medium was replaced with polymer extracts or solutions of initiators and the cells were cultured for the next 24 h. As an indicator of cellular death, the release of lactate dehydrogenase (LDH) was determined using the In Vitro Toxicology Assay Kit, Lactate Dehydrogenase Based (Sigma Aldrich). In selected wells, cells were lysed before the assay to determine the maximal LDH release (high control). Then, plates were centrifuged (200 RCF, 4 min.) and medium samples of 50 µL each were transferred to the separate test microplates. LDH activity in the media was assessed according to the manufacturer’s instruction. The absorbance was measured at wavelengths of 490 nm and 690 nm (background) and cytotoxicity was calculated according to the following formula: Cytotoxicity (%) = (A_T_ − A_C_/A_H_ − A_C_)∗100(1)
where A_T_ is the absorbance of the treated wells; A_C_ is the control; and A_H_ is the high control (maximal LDH release). 

In the plates with cells, the culture medium was replaced with a fresh one, and the viability of the cells was assessed using the In Vitro Toxicology Assay Kit, Resazurin Based (Sigma Aldrich). Cells were incubated with the resazurin solution and the absorbance was measured at wavelengths of 595 nm and 690 nm (background). Absorbance decrease relative to the blank (wells without cells) was used as an indicator of cell viability. 

### 3.7. Measurements

The conversion of the reaction and structure of the obtained products was determined with NMR spectroscopy. The ^1^H NMR spectra were recorded at 600 MHz with a Bruker Avance IITM 500 MHz at 25 °C. Dried DMSO-d6 was used as a solvent, and tetramethylsilane was applied as the internal standard. The spectra were obtained with 64 scans, a 2.65 s acquisition time, and a 11 µs pulse width. 

The number-average and weight-average molar masses of the oligomers were determined by gel permeation chromatography with a Viscotek RImax chromatograph (Malvern Panalytical Ltd., Malvern, UK). Chloroform was used as the eluent, and the temperature and flow rate were 35 °C and 1 mL/min, respectively. Two PL Mixed E columns with a Viscotek model 3580 refractive index detector and injection volume equal to 100 mL were used.

FTIR spectra were recorded on a JASCO FTIR-6700 (JASCO Deutschland GmbH, Pfungstadt, Germany) spectrometer, using a TGS detector with 64 scans and at 4 cm^−1^ resolution. Samples were analyzed in a form of pellets in KBr. 

The percentage of carbon, hydrogen, and nitrogen in the samples was determined by the VARIO EL III Element analyzer (Elementar Analysensysteme GmbH).

### 3.8. DFT Calculations 

All geometric structures of the zinc complexes were fully optimized at the B3LYP/6-311G* density functional (DFT) level [48] by using the Gaussian 03 Rev. E.01-SMP program [49]. The obtained geometry was visualized with the GaussView 4.1 program [50]. 

## 4. Conclusions

By using carefully selected ROP initiators of lactides with good polymerization initiation efficiency, low toxicity, and high antibacterial activity, it is possible to obtain a bacteriostatic polymer with properties relevant for many biomedical applications. In the present work, the use of zinc complexes allowed to obtain some aliphatic bioresorbable co-polyesters with the ability to inhibit the growth of bacteria and fungi and showing biocompatibility with skin cells. The types of oligo- and polyesters obtained with Zn[(acac)(LPhe)H_2_O], in the form of special carriers of the bioactive agents, as an ingredient for formulations, could be used in cosmetology and dermatology. 

## Figures and Tables

**Figure 1 ijms-22-06950-f001:**
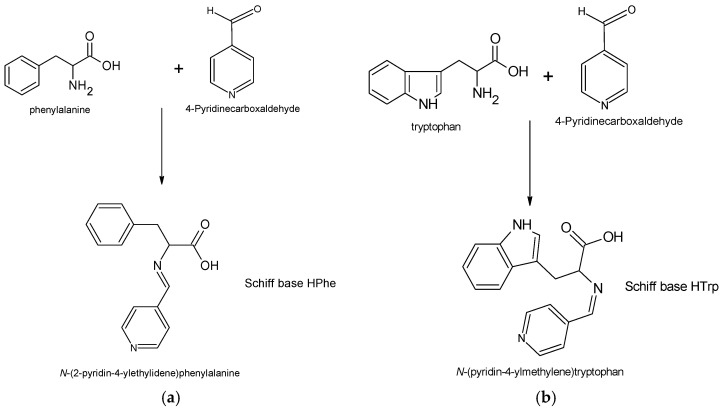
Synthesis pathway and structure of the Schiff base ligands: (**a**) HPhe obtained with l-phenylalanine, and (**b**) HTrp obtained with tryptophan.

**Figure 2 ijms-22-06950-f002:**
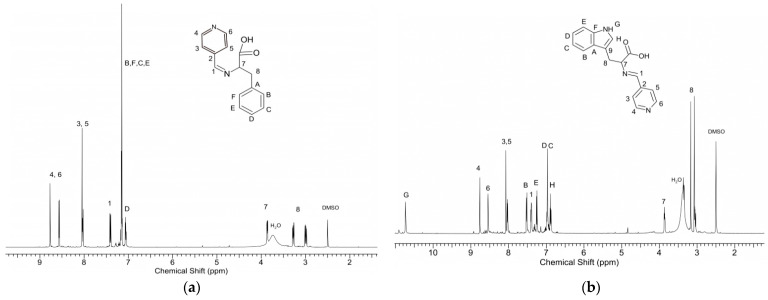
^1^H NMR spectra (in DMSO d_6_) of the Schiff base; HPhe obtained with l-phenylalanine (**a**), and HTrp obtained with tryptophan (**b**).

**Figure 3 ijms-22-06950-f003:**
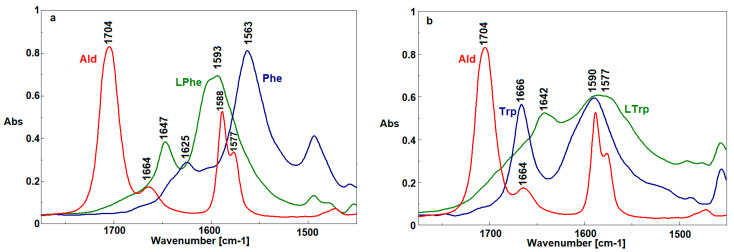
FTIR spectra of HPhe, 2-C-pyridine aldehyde, and phenylenealanine (**a**), and HTrp, 2-C-pyridine aldehyde and tryptophane (**b**).

**Figure 4 ijms-22-06950-f004:**
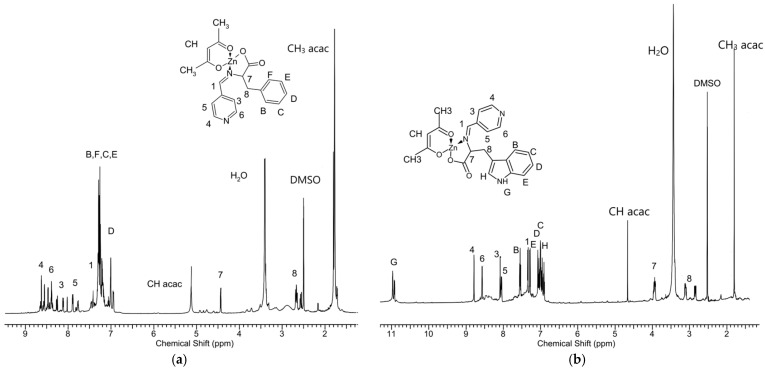
^1^H NMR spectra (in DMSO d_6_) of (**a**) Zn[(acac)(Lphe)H_2_O], and (**b**) Zn[(acac)(LTrp)H_2_O].

**Figure 5 ijms-22-06950-f005:**
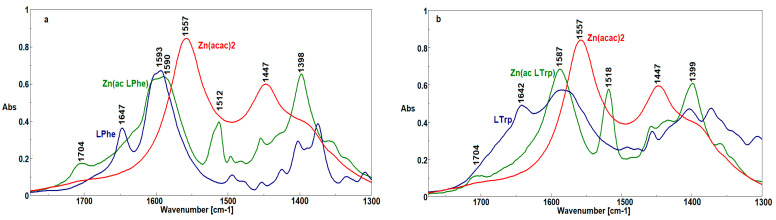
FTIR spectra of Zn[(acac)(LPhe)H_2_O], Zn[(acac)_2_ H_2_O], and HPhe (**a**), and Zn[(acac)(LTrp) H_2_O], Zn(acac)_2_ H_2_O], and HTrp (**b**).

**Figure 6 ijms-22-06950-f006:**
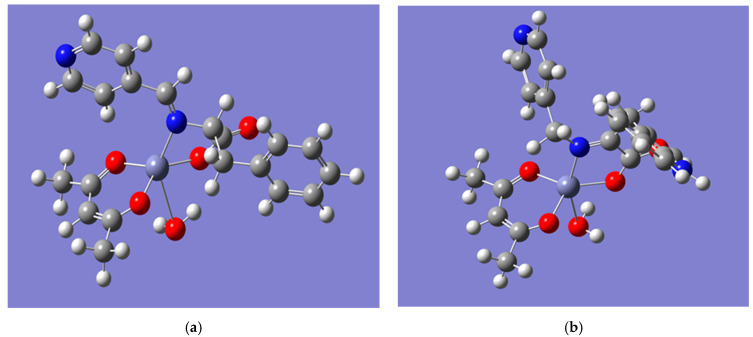
The geometry of the complexes calculated by Gaussian 03 with the DFM method and B3LYP/6-31G^∗^ Orbital Basis Set: (**a**) Zn [(acac) (LPhe)H_2_O], and (**b**) Zn[(acac)(LTrp)H_2_O].

**Figure 7 ijms-22-06950-f007:**
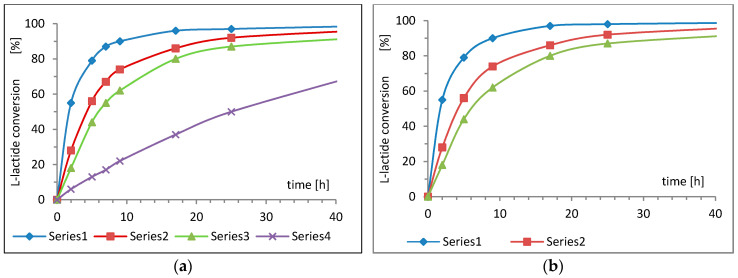
The dependence of lactide conversion and time of the conducted polymerization in bulk at 120 °C, initiated by (**a**) Zn[(acac)(LPhe)H_2_O], and (**b**) Zn[(acac)(LTrp)H_2_O]. Series 1—with an M/I ratio of 150:1; Series 2—with an M/I ratio of 400:1; Series 3—with an M/I ratio of 600:1; Series 4—Zn[(acac)H_2_O] with an M/I ratio of 600:1.

**Figure 8 ijms-22-06950-f008:**
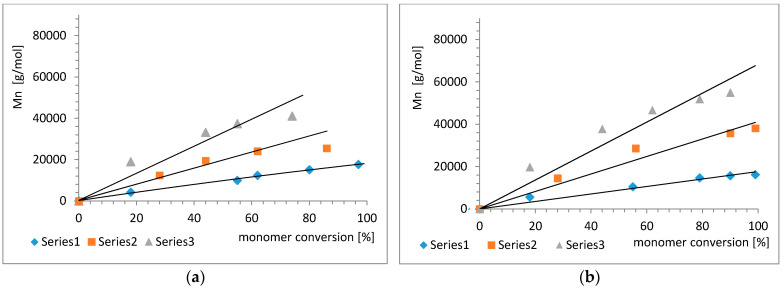
Relationship between the average molecular weight of the polylactide and monomer conversion; polymerization was conducted in bulk at 120 °C, initiated by Zn[(LPhe)(acac)H_2_O] (**a**), and Zn[(acac)(LTrp)H_2_O] (**b**). Series 1—with an M/I ratio of 150:1; Series 2—with an M/I ratio of 400:1; and Series 3—with an M/I ratio of 600:1.

**Figure 9 ijms-22-06950-f009:**
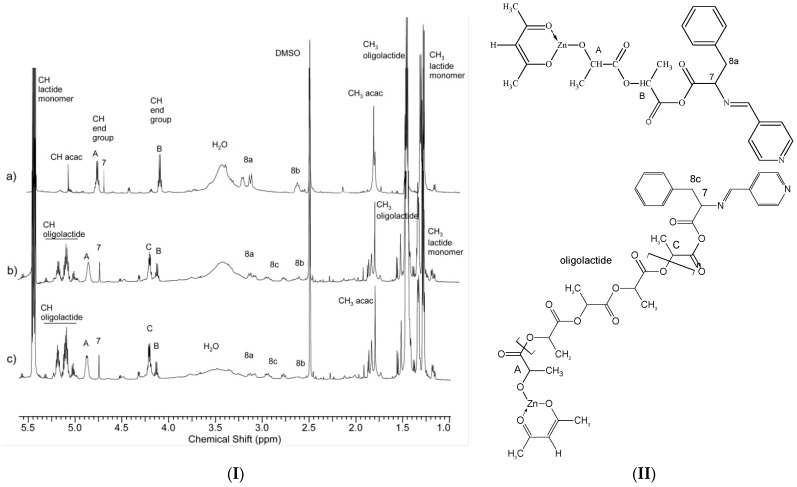
(**I**) ^1^H NMR spectra of the product of the l-lactide polymerization conducted in benzene with Zn[(acac)(Lphe)]xH_2_O as an initiator with an I/M ratio of 1:12: after 20 min (**a**), 1.5 h (**b**), and 9 h (**c**). (**II**) Hypothetical chemically dominated structure of the products of l-lactide polymerization with use of Zn[(acac)(Lphe)H_2_O] as the initiator: after 20 min. (**a**,**b**).

**Figure 10 ijms-22-06950-f010:**
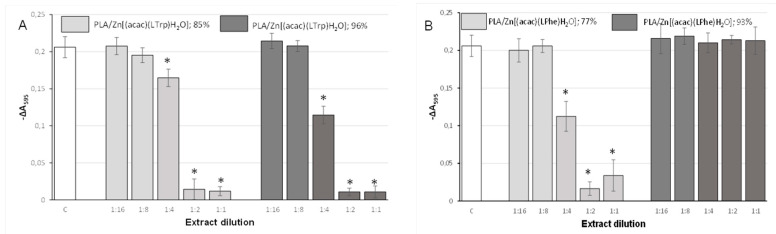
Resazurin reduction by CCD-11Lu fibroblasts incubated with extracts of polylactides initiated using (**A**) Zn[(acac)(LTrp)H_2_O]; and (**B**) Zn[(acac)(LPhe)H_2_O]. Percentages describe the total conversion rate, and each bar represents the mean ± SD; * *p* < 0.05 compared to the control.

**Figure 11 ijms-22-06950-f011:**
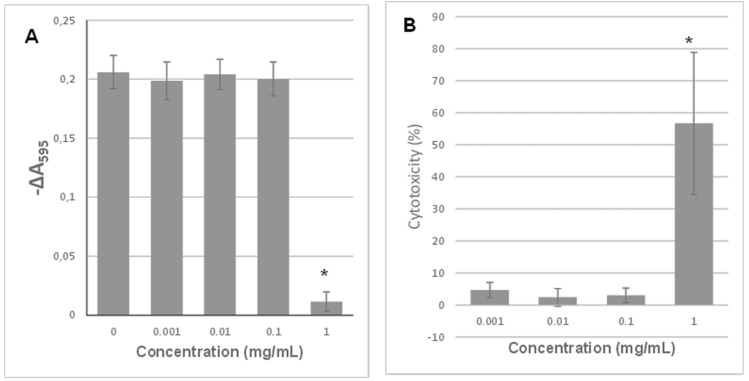
Effect of 24 h exposure to Zn[(acac)(LPhe)H_2_O] on (**A**) the resazurin reduction by CCD-11Lu cells; (**B**) LDH release from CCD-11Lu cells. Each bar represents the mean ± SD; * *p* < 0.05 compared to the control.

**Table 1 ijms-22-06950-t001:** Screening of the effect of the present selected compounds on the inhibition of the growth of bacterial and fungal strains (own research results). Approximate values of the minimum inhibitory concentration (MIC).

Compound		Growth Inhibition Concentration (mg/mL)					
		*Escherichia coli*	*Pseudomonas aeruginosa*	*Staphylococcus epidermidis*	*Staphylococcus aureus*	*Candida albicans*	*Aspergillus brasiliensis*
1	Zn[(acac)_2_ H_2_O]	1 after 24 and 48 h average	10 after 24 and 48 h average	10 after 24 and 48 h average	1 after 24 and 48 h average	1 after 24 and 48 h average	10 after 24 and 48 h average
2	Spermidine	10 after 24 and 48 h average	10 after 24 and 48 h average	1 after 24 and 48 h average	1 after 24 and 48 h total	1 after 24 and 48 h total	10 after 24 and 48 h average
3	Putrescine	10 after 24 and 48 h total	10 after 24 and 48 h total	10 after 24 and 48 h Total	10 after 24 and 48 h total	10 after 24 and 48 h total	10 after 24 and 48 h average
4.	Phenylalanine	10 > not observed	10 after 24 and 48 h minimal	10 after 24 and 48 h minimal	10 after 24 and 48 h minimal	10 after 24 and 48 h minimal	10 after 24 and 48 h minimal
5	Schiff base HTrp	10 total after 48 h	10 after 24 h average	1 after 24 and 48 h average	10 total after 48 h average	1 after 24 and 48 h average	1 after 24 and 48 h average
6	Schiff base HPhe	10 after 24 and 48 h average	10 after 24 and 48 h average	1 after 24 and 48 h average	10 after 24 and 48 h average	10 after 24 and 48 h average	1 after 24 and 48 h average
7	Zn[(acac)(LTrp)H_2_ O]	0.1 after 48 h average	1 after 24 and 48 h average 0.1 minimal	0.1 after 24 and 48 h average	1 after 24 and 48 h average	0.1 after 24 and 48 h average	10 > minimal
8	Zn[(acac)(LPhe)H_2_O]	0.1 after 24 and 48 h average	1 after 24 h total 0.1 after 48 h total	0.1 after 24 h and 48 h average	0.1 after 24 and 48 h average	1 after 24 and 48 h average	1 after 48 total

Note: after 24 and 48 h—after 24 h and 48 h of incubation; minimal—decrease in the number of cells by no more than 50%; average—decrease in the number of cells from 50% to 90%; total—decrease in the number of cells by more than 99.9%, compared to the number of cells in the control sample.

**Table 2 ijms-22-06950-t002:** Screening of the effect of the presence of selected samples of poly(l-lactide) on the inhibition of the growth of bacteria and fungi strains.

Poly(l-Lactide) Initiator Molar Ratio I:M		Growth Inhibition Concentration (mg/mL)					
		*Escherichia coli*	*Pseudomonas aeruginosa*	*Staphylococcus epidermidis*	*Staphylococcus aureus*	*Candida albicans*	*Aspergillus brasiliensis*
1	Tin (II) 2-ethylhexanoate 1:400	1 after 48 minimal	10 > not observed	10 > after 24, 48 h minimal	10 > after 24, 48 h minimal	1 after 24, 48 h minimal	10 > not observed
2	Zn[(acac)(LPhe)H_2_O] 1:150	0.1 after 24, 48 h total	0.1 after 24 h average 0.01 after48 h minimal	0.1 after 24, 48 h average	0.1 after 24, 48 h average 0.01 after 48 minimal	10 > after 24, 48 h minimal	0.1 after 24 h,48 h total 0.01 After 48 h minimal
3	Zn[(acac)(LPhe)H_2_O] 1:400	1 after 24 h average 0.1 after 48 h average	1 after 24 h total 0.1 after 48 h minimal	0.1 after 24 h minimal 0.1 after 48 average	0.1 after 24, 48 h average 0.01 after 24, 48 h not observed	10 > after 24, 48 h minimal	1 after 48 h average 0.1 after 48 h minimal
4	Zn[(acac)(LTrp)H_2_O] 1:150	10 after 24 h minimal 1 after 48 h average	1 after 24 h average 10 > after 48 h minimal	0.1 after 24, 48 h average	0.1 after 24 h average 10 after 48 minimal	0.1 after 24 average 10 after 48 h minimal	10 > not observed
5	Zn[(acac)(LTrp)H_2_O] 1:400	10 after 24 h minimal 1 after 48 h average	10 after 24 h average 10 > after 48 h minimal	0.1 after 24 h, 48 h minimal	0.1 after 24 h minimal after 48 h not observed	1 after 24 h minimal after 48 h not observed	10 > not observed

Note: after 24 and 48 h—after 24 h and 48 h of incubation; minimal—decrease in the number of cells by no more than 50%; average—decrease in the number of cells from 50% to 90%; total—decrease in the number of cells by more than 99.9%, compared to the number of cells in the control sample.

**Table 3 ijms-22-06950-t003:** Properties of the tested polymers.

Initiator Molar Ratio I/M	Total Conversion (%)	M_n_ (g/mol)	Ð
Tin (II) 2-ethylhexanoate 1: 400	93	50,200	2.3
Zn[(acac)(LPhe)H_2_O] 1:150 Zn[(acac)(LPhe)H_2_O] 1: 400	97	10,200	2.8
	95	31,700	2.2
Zn[(acac)(LTrp)H_2_O] 1:150	98	12,100	2.6
Zn[(acac)(LTrp)H_2_O] 1:400	98	38,900	2.1

Note: Poly(l-lactide) obtained in bulk, at 120 °C, after 36 h. I/M—molecular ratio initiator: l-lactide; Mn—the average molecular mass; Ð—dispersion of the molecular mass.

## Data Availability

Most of the data was included in the publication. The others source data presented in this study are available on request from the corresponding author. The data are not publicly available due to the present lack of access to a trusted public depository.

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
