# Peer review of "Synthesis of the Bacteriostatic Poly(l-Lactide) by Using Zinc (II)[(acac)(L)H_2_O] (L = Aminoacid-Based Chelate Ligands) as an Effective ROP Initiator"

_ijms, 2021, doi:10.3390/ijms22136950_

Round 1

Reviewer 1 Report

This paper reports the synthesis of two new Zn(II) complexes in which the metal is coordinated by an acac ligand and a tryptophane or phenylalanine-containing  Schiff base. The authors then analyse their antibacterial and antifungal activity. 

The subject of the present is undoubtedly of large interest considering the relevance of the development of new antibacterial agents against important bacterial strains, such as staphylococcus aureus. The paper is easy to read and the author conclusions are adequately supported by the experimental results.  On the light of this consideration, I support publication of this manuscript.  My only criticism is the of any experiment to analyse the toxicity of the present complexes. This point should be clarified before publication of this interesting manuscript.

Author Response

Reviewer 1.

The subject of the present is undoubtedly of large interest considering the relevance of the development of new antibacterial agents against important bacterial strains, such as staphylococcus aureus. The paper is easy to read and the author conclusions are adequately supported by the experimental results.  On the light of this consideration, I support publication of this manuscript.

Thank you very much and we are pleased with such a good evaluation of our work. We fully agree that the subject of works on methods of combating the threat posed by bacteria and fungi is very topical. Conducting such research on a large scale is becoming even necessary.

My only criticism is of any experiment to analyse the toxicity of the present complexes. This point should be clarified before publication of this interesting manuscript.

Thank you for your very apt advice. Indeed, the evidence of low toxicity of the initiators used, as well as of the final polymer, was very much lacking in the previous version of our work.
Therefore, we performed additional in vitro cytotoxicity studies on both the zinc complexes and the final polylactide obtained with these compounds. It took some time. We have devoted a separate chapter to this topic. We showed that both the Zn [(acac) (Lphe) H2O] complex and the polymer obtained with this compound practically not show toxicity to the selected test cell line.  We also investigated the cytotoxicity of the polymer obtained with use of Zn[(acac) (LTrp) H2O], which turned out to be only slightly higher. We also investigated the influence of unreacted monomer (lactide) on the cytotoxicity of the tested polymerization product. The research was carried out by a friendly team from the Silesian Medical University.

Reviewer 2 Report

The manuscript presents an interesting work on the synthesis of the antibacterial metallocomplex. Overall the language of the manuscript requires significant improvement since some phrases are very difficult to understand. Among a few examples, without going too much into details, * "Interesting from this point of view regarded zinc 84 complexes, fully meeting the expected requirements; presenting robust antibacterial activity and relatively low toxicity", "elementary analysis" (correct term is elemental analysis) etc. I strongly suggest to either acquire language services or ask a native English speaker to read through the text.

Table with screening data looks confusing. Did the authors screen all these compounds themselves or had some literature search been performed. If screening involved also the literature search, references should be added into the table, in a separate column, for example.

Methodologically structure optimization carried out in HyperChem does not sound to be performed on the good level of stringency. Typically HyperChem is used for preliminary energy minimization of the structure before loading it into PC-GAMESS, Quantum Espresso, Priroda, or similar. I would suggest either replacing that part of the text with some more reliable simulations or perform X-ray structural analysis, which would be the most preferable way to learn something about the complexes, or removing HyperChem simulations completely.

Figure 8 data are fitted with linear functions whereas there is no such dependency, especially at high monomer conversions. I suggest removing linear fits to avoid confusion.

For polymerization and viability studies, it looks like controls are completely missing.

Author Response

Reviewer 2.

The manuscript presents an interesting work on the synthesis of the antibacterial metallocomplex. Overall the language of the manuscript requires significant improvement since some phrases are very difficult to understand. Among a few examples, without going too much into details, * "Interesting from this point of view regarded zinc 84 complexes, fully meeting the expected requirements; presenting robust antibacterial activity and relatively low toxicity", "elementary analysis" (correct term is elemental analysis) etc. I strongly suggest to either acquire language services or ask a native English speaker to read through the text.

In order to improve grammar, vocabulary and clarity of the manuscript, this text has been subjected to correction by a professional certified company specializing in translation and language service. (https://translmed.com/korekta-jezykowa/)

Table with screening data looks confusing. Did the authors screen all these compounds themselves or had some literature search been performed. If screening involved also the literature search, references should be added into the table, in a separate column, for example.

All the data in the table are the results of our work. Indeed this was unclear. So we introduced an appropriate explanation in the description of the table, and also indicated this fact in the text of the manuscript.

Methodologically structure optimization carried out in HyperChem does not sound to be performed on the good level of stringency. Typically HyperChem is used for preliminary energy minimization of the structure before loading it into PC-GAMESS, Quantum Espresso, Priroda, or similar. I would suggest either replacing that part of the text with some more reliable simulations or perform X-ray structural analysis, which would be the most preferable way to learn something about the complexes, or removing HyperChem simulations completely.

We agree with the opinion of the reviewer. We recalculated the geometry of the complexes using the suggested Gaussian program. All geometric structures of the zinc complexes were fully optimized at the B3LYP / 6-311G * density functional (DFT) level by using the Gaussian 03 Rev. E.01-SMP program.

We inserted the obtained results into the manuscript (additionally files with a 3D image in the supplement). We have not observed the fundamental changes in the structure of the geometry in comparison to the results obtained with the HyperChem program. Structure testing with X-ray structural analysis was not possible, as we were unable to obtain the appropriate monocrystals.

Figure 8 data are fitted with linear functions whereas there is no such dependency, especially at high monomer conversions. I suggest removing linear fits to avoid confusion.

The linear functions presented in the previous version of the manuscript pictured theoretical relationships between monomer conversion and the number average molecular weight. As suggested, we removed them from the figure and replaced them with the trend function of the obtained experimental data (Figure 8). We have also deleted the part describing how to calculate the theoretical molecular weight values, which is not needed in this case.

Thank you very much for a basically favorable and constructive review.

Round 2

Reviewer 2 Report

I am happy to recommend the manuscript for publication